**Subject Category:**
Biology (whole organism)

plant science/chemical biology/biotechnology

*Ageratina adenophora*, oxidative stress, reactive oxygen species, Nod-like receptor protein 3, pyroptosis

**Author for correspondence:**
Yanchun Hu
e-mail: hychun114@163.com

†Those authors contributed equally to this work.

# *Ageratina adenophora* causes spleen toxicity by inducing oxidative stress and pyroptosis in mice

Wei Sun[1,2,†], Chaorong Zeng[3,†], Dong Yue[1,†],
Shanshan Liu[2,†], Zhihua Ren[1], Zhicai Zuo[1],
Junliang Deng[1], Guangneng Peng[1] and Yanchun Hu[1]

[1]Key Laboratory of Animal Disease and Environmental Hazards of Sichuan Province, College of Veterinary Medicine, Sichuan Agricultural University, Wenjiang District, Chengdu, Sichuan 611130, People's Republic of China
[2]Tongren Polytechnic College, Bijiang District, Tongren, Guizhou 554300, People's Republic of China
[3]Affiliated Sichuan Provincial Rehabilitation Hospital of Chengdu University of TCM, Sichuan Bayi Rehabilitation Center, Chengdu, Sichuan 611135, People's Republic of China

 YH, 0000-0002-2998-7879

*Ageratina adenophora* is an invasive weed with potent toxicological effects on livestock. Oxidative stress and pyroptosis play a pivotal role in regulating animal or human health and disease. The object of this study was to determine the mechanism underlying splenic toxicity induced by *A. adenophora* in a mouse model. *Ageratina adenophora* significantly increased the levels of reactive oxygen species and malondialdehyde, but decreased the antioxidants like catalase, superoxide dismutase, glutathione and glutathione peroxidase. In addition, the activity of the antioxidant enzymes was also decreased upon *A. adenophora* treatment. The induction of the pyroptosis pathway was evaluated in terms of the expression levels of Nod-like receptor protein 3, nuclear factor-κB, caspase-1, gasdermin-D and interleukin-1β, all of which were significantly elevated by *A. adenophora*. These findings suggest that *A. adenophora* impairs spleen function in mice through oxidative stress damage and pyroptosis.

## 1. Introduction

*Ageratina adenophora*, also known as *Eupatorium adenophorum*, is an invasive weed species native to Mexico and Costa Rica and has successfully invaded habitats across Europe, Oceania and Asia [1]. The southwest provinces of China are one of the worst affected regions, where *A. adenophora* reduces the biomass of other

plants by altering the soil microbial community in the invaded areas [2,3]. The invasion of *A. adenophora* to grassland indirectly leads to the reduction of the number of grazing animals and local plants and the loss of biodiversity. In addition, *A. adenophora* is highly toxic to animals and affects multiple organs. For example, ingestion of this weed causes respiratory disease in horses [4]. Intragastric administration of the freeze-dried leaf powder or methanol extract of *A. adenophora* resulted in multiple focal parenchymal necrosis and liver degeneration in mice [5,6]. Rats fed with chow containing 25% (w/w) freeze-dried *A. adenophora* leaf powder developed jaundice, characterized by increased levels of plasma bilirubin, ALP, ALT and AST [7,8]. Furthermore, rumination suspension and photosensitization have been caused in cattle [9]. The toxic effects of *A. adenophora* ingestion on the liver [10], spleen [11] and kidney [12] of goat have also been demonstrated, with dose-dependent apoptosis and autophagy seen in goat tissues. A study has demonstrated that *A. adenophora* induced significant mice oxidative stress characterized by upregulating mRNA levels of antioxidants, including superoxide dismutase (SOD), catalase (CAT) and glutathione (GSH) [13]. Consistent with this, our previous study has proved that greater than or equal to 20% dose of *A. adenophora* increased the liver weight and caused extensive inflammation, in addition to decreasing antioxidant activity, increasing the levels of reactive oxygen species (ROS) [14].

Oxidative stress is defined as the imbalance between the production and scavenging of ROS due to impaired antioxidant defence mechanisms, which results in a net increase in ROS. Endogenous ROS are products of normal cellular metabolism and can be scavenged by antioxidant enzymes like CAT, SOD, GSH and glutathione peroxidase (GPx) [15,16]. However, overproduction of ROS overwhelms the antioxidant system and leads to oxidative stress that damages lipids, proteins and nucleic acids, and even causes cell death [17,18]. Increased levels of ROS are associated with a variety of diseases, such as chronic inflammation [19], and promote the release of various pro-inflammatory factors [20]. Pyroptosis is a new form of programmed cell death that is accompanied by an inflammatory response involving both caspase-1 activation and interleukin-1β (IL-1β) production [21–26]. The caspase-1 protease is a core component of multiprotein inflammasome complexes that trigger the secretion of IL-1β, a pro-inflammatory cytokine [27]. Studies also show that pyroptosis is an immune effector mechanism in several cell types [24,25] and is triggered by diverse pathological stimuli [28,29], leading to the secretion of pro-inflammatory cytokines and intracellular contents [30]. Although a previous report has indicated that *A. adenophora* can induce an extensive inflammatory response [31], the underlying mechanism is still unclarified.

As the largest peripheral lymphatic organ, the spleen contains around one-fourth of the body's lymphocytes and plays a vital role in initiating an immune response [32,33]. It is known that *A. adenophora* can induce varying degrees of histopathological damage to the spleen [6], indicating that its toxic effects probably involved the immune system. In addition, *A. adenophora* administration into Kunming mice disrupted the splenocytic arrangement and natural killer cells activity [34]. One study also showed a direct role of *A. adenophora* in DNA damage and apoptosis in the spleen [11].

However, apoptosis is a form of cell death that avoids causing inflammation. This discrepancy might indicate a new cell death involved in the process of inflammation caused by *A. adenophora*. Pyroptosis is a pro-inflammatory form of regulated cell death, which is regarded as a general immune effector in multiple cells. Although some previous studies have investigated the mechanism of toxicity induced by *A. adenophora*, the role of pyroptosis and oxidative stress in the damage caused by *A. adenophora* is still unknown. The aim of this study was to determine the possible mechanism of *A. adenophora*-induced oxidative stress and pyroptosis in the murine spleen. Briefly, we measured levels of ROS and malondialdehyde (MDA), along with that of antioxidant enzymes and pyroptosis regulators, such as NLRP3, NF-κB, caspase-1, Gasdermin D (GSDMD) and IL-1β. Our findings may provide a new insight into the mechanism of splenic damage and toxicity induced by *A. adenophora*.

# 2. Material and methods

## 2.1. Reagents

Total ROS Assay kit 520 nm (No. 88-5930-74) was purchased from Thermo Fisher, USA. Biochemical assay kits were obtained from Nanjing Jiancheng Bioengineering Institute (Nanjing, Jiangsu, China). Apoptosis Kit (No. 556547) was purchased from Becton Dickinson Company (Franklin Lakes, USA). Antibodies against Nod-like receptor protein 3(NLRP3) and nuclear factor-κB (NF-κB) were obtained from Sagon Biotech (Shanghai, China). The antibody against Gasdermin D was obtained from Thermo Fisher (USA) and β-actin from Bioss (Beijing, China). Caspase-1 p20 antibody was purchased from

Boster Biological Technology Co., Ltd (Wuhan, Hubei, China). *Ageratina adenophora* was collected from Xichang, Sichuan Province, Southwest China. The collected leaves were cleaned, ground and screened at room temperature to make dry powder. The powder was stored in shade condition with an ambient temperature of $20 \pm 2°C$. For the preparation of 10%, 20% and 30% diet pellet, *A. adenophora* and mice feed were homogenized in water solution by the ratio of $1:9$, $1:8$ and $1:7$, respectively. Then the diet was cast in the form of cylinders and dried at 27°C for 48 h.

## 2.2. Experimental animals

Forty 8-week-old female Kunming mice bought from Experimental Animal Corporation of Dossy Biological Technology Company (Chengdu, Sichuan, China) were divided randomly into control group and groups A–C ($n = 10$). The mice in the control group were fed with a nutritionally balanced diet. Groups A–C were administrated with 10%, 20% and 30% dosage of *A. adenophora* pellet diet, respectively. Water was provided ad libitum.

## 2.3. Assay of oxidative stress-related indexes of the spleen

After 42 days of experimentation, mice in all groups were anaesthetized and sacrificed, and the spleens were immediately removed and washed with phosphate buffered saline (PBS, pH 7.4). The spleens were homogenized with nine volumes pre-cooled normal saline buffer in an ice bath and centrifuged at 4°C for 10 min at 4000 r.p.m. to obtain a supernatant. The supernatants were collected to measure the contents of GSH and MDA, as well as the activities of SOD, CAT and GPx through corresponding kits (Nanjing Jiancheng Bioengineering Institute, Nanjing, Jiangsu, China), after determining the protein content by a 'Modified Bradford Protein Assay Kit' (Sagon Biotech, Shanghai, China).

## 2.4. Reactive oxygen species analysis by flow cytometry

At the end of the experiment, the spleen was ground and filtered with a 300-mesh nylon membrane to obtain splenocytes. Splenocyte with a concentration of $1.0 \times 10^6$ cells $ml^{-1}$ was stained with $1 \times ROS$ assay solution and incubated for 20 min at 37°C in the dark room and then the production of ROS was measured by a flow cytometer (Becton Dickinson, USA).

## 2.5. Detection of pyroptosis by flow cytometry

Mice were sacrificed humanely at 42 days, and the spleen was separated from mouse and grounded to obtain suspension, filtered with a 300-mesh nylon screen to obtain splenocytes and then washed three times with PBS. Cell pellets were resuspended in PBS and incubated with annexin V-FITC/PI apoptosis staining at room temperature for 15 min in the dark room. Then, 300 µl binding buffer was added. Quantitative analysis of pyroptosis cells was detected with a flow cytometer (Becton Dickinson, USA), and the data were visualized by CellQuest Pro software (Becton Dickinson, USA).

## 2.6. Immunohistochemistry assay

The spleen was carefully dissected from the mice, washed with cold PBS (pH 7.2–7.4) and then fixed overnight in 4% paraformaldehyde, and embedded in paraffin wax after dehydration. The paraffin-embedded spleen tissue was sliced into 5 µm sections and then dewaxed in xylene, followed by rehydrating through a graded series of ethanol solutions. Endogenous peroxidase was blocked by incubating with 3% $H_2O_2$ in methanol for 15 min, and heat-induced antigen retrieval using Tris–EDTA (pH 8) was performed. The sections were sealed with 5% normal serum and incubated overnight at 4°C with the primary antibody. The slices were exposed to a biotinylated secondary antibody for 1 h. Immunoreaction was visualized by using DAB as a substrate.

## 2.7. The caspase-1 activity assay

The activity of caspase-1 in each group was determined by using a commercial assay kit (Beyotime, Shanghai China) according to the operational manual. In brief, 20 mg spleen tissue was lysed on ice in 200 µl for 30 min. The protein concentration in the supernatants was measured through BAC method (Sagon Biotech, Shanghai, China). The caspase-1 protease activity was measured at 405 nm

with a microplate spectrophotometer (Bio-Tek Instruments, Inc., USA), following by incubation with the substrate at 37°C for 2 h.

## 2.8. Enzyme-linked immunosorbent assay

After 42 days of experimentation, mice in all groups were euthanized, and 10% spleens supernatant was obtained by the above-mentioned method. IL-1β concentration was measured through enzyme-linked immunosorbent assay (ELISA) method (Wuhan, China) as the protocol provided. Briefly, 100 µl sample was added into the plates and incubated at 37°C for 1 h. After washing thoroughly with PBST (PBS + 0.05% Tween 20), the plates were incubated with 100 µl HRP-labelled working solution at 37°C for 30 min in the dark and washed again. The reaction was terminated with 50 µl 2 mol l$^{-1}$ $H_2SO_4$ solution and the optical density was measured at 450 nm by a microplate reader.

## 2.9. RNA extraction and quantitative real-time PCR analysis

The spleen in each group was collected and ground into powder by liquid nitrogen in a mortar. Total RNA was isolated from spleens using lysis buffer by an 'Animal Total RNA Isolation Kit' (Sagon, Shanghai, China) following the instructions provided. PrimeScript RT reagent kit (TAKARA, Japan) was used for the reverse transcription of RNA into cDNA for qRT-PCR. The first-strand cDNA was amplified using SYBR on LightCycler 96 (Roche, Germany). The primers were designed using Oligo7 software and synthesized by Sagon Biotech Ltd (Shanghai, China) (electronic supplementary material, table S1). The housekeeping gene β-actin was introduced as an internal control for data normalization. The data analysis was performed with $2^{-\Delta\Delta}$ Ct method.

## 2.10. Western blotting

Spleen tissues were lysed and proteins were extracted through a 'Tissue or Cell Total Protein Extraction Kit' (Sagon, Shanghai, China). Protein concentration in each group was detected by a 'Modified Bradford Protein Assay Kit' (Sagon Biotech, Shanghai, China). The protein samples were separated by 10% sodium dodecyl sulphate polyacrylamide gel electrophoresis and transferred onto a polyvinylidene difluoride membrane (PVDF). The membrane was blocked with a 5% non-fat milk solution at room temperature for 90 min and then incubated overnight with targeted primary antibodies at 4°C with gentle shaking. After washing three times with PBST buffer, the PVDF membrane was incubated with the corresponding secondary antibody at room temperature for 1 h. Blots were visualized by DAB (Sagon Biotech, Shanghai, China) followed by washing three times with PBST. β-actin was introduced as a control.

## 2.11. Statistical data analysis

The data are represented as means ± s.d. One-way analysis of variance (ANOVA) SPSS 22.0 software package (SPSS Inc., USA) was used to assess statistical significances between *A. adenophora*-treated groups and the control group. All statistical artworks were performed by GraphPad Prism 6.0. A value of $p < 0.05$ or $p < 0.01$ was regarded as a significant difference.

# 3. Results

## 3.1. Changes of oxidative damage indices in the spleen

All *A. adenophora*-treated groups showed a marked decrease in the splenic activity of CAT (figure 1*a*, $p < 0.01$), GPx (figure 1*b*, $p < 0.01$) and Mn-SOD (figure 1*c*, $p < 0.01$) compared between groups in a dose-dependent manner. Interestingly, Cu–Zn SOD activity was only decreased in groups B and C compared with the control group, but not in group A (figure 1*d*). *Ageratina adenophora* treatment also resulted in a significant decrease in the level of GSH (figure 1*e*, $p < 0.01$) and increase in MDA content (figure 1*f*; $p < 0.05$, $p < 0.01$) by comparing between all treated groups.

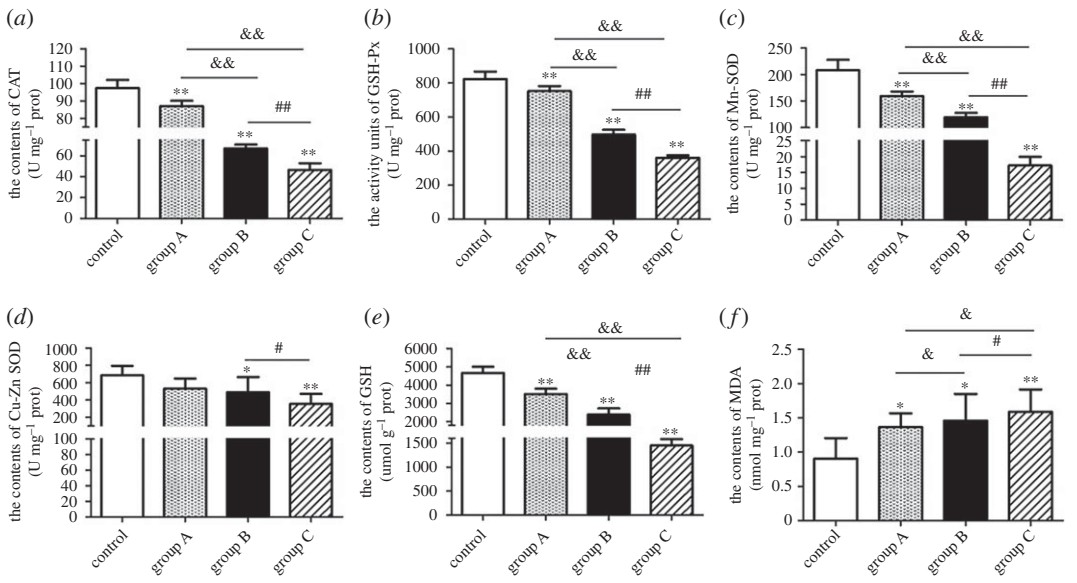

**Figure 1.** Oxidative stress markers in the murine spleen following *A. adenophora* administration. (*a*) CAT, (*b*) GPx, (*c*,*d*) SOD, Mn-SOD and Cu–Zn SOD, (*e*) GSH and (*f*) MDA protein levels. The data are presented with the mean ± s.d., $^*p < 0.05$ and $^{**}p < 0.01$ versus the control group; $^{\&}p < 0.05$ and $^{\&\&}p < 0.01$ versus group A; $^{\#}p < 0.05$ and $^{\#\#}p < 0.01$ versus group B.

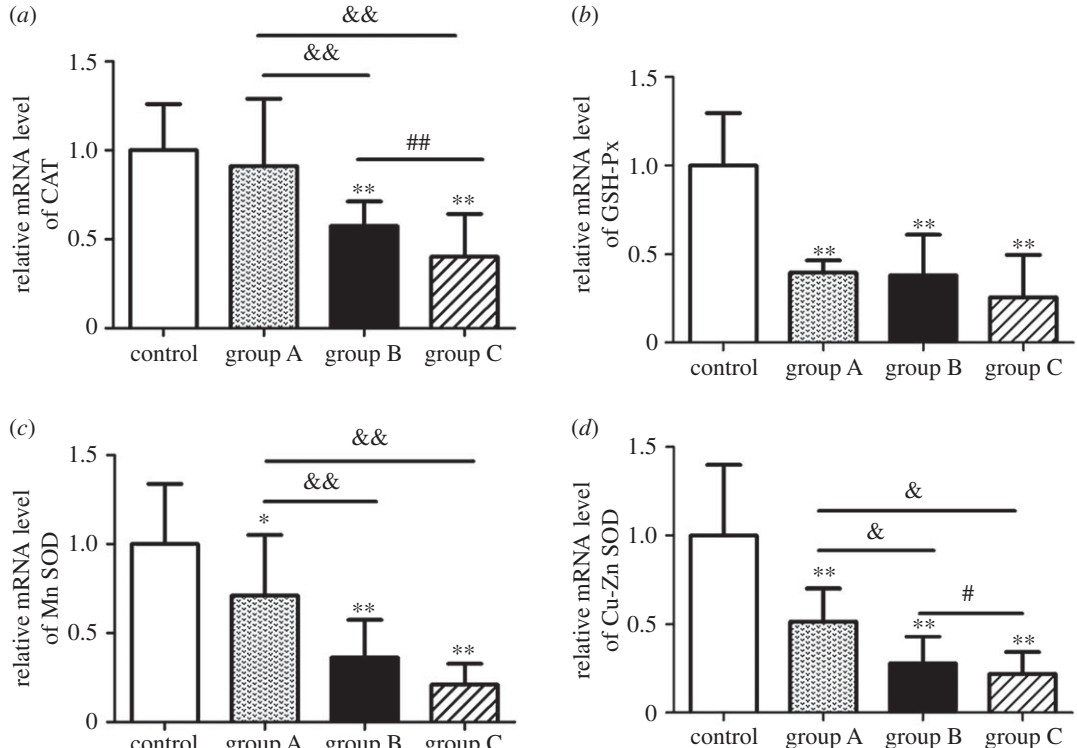

**Figure 2.** *Ageratina adenophora* transcriptionally downregulated the antioxidant enzymes. (*a*) CAT, (*b*) GPx, (*c*,*d*) total SOD, Mn-SOD and Cu–Zn SOD mRNA levels. The data are represented as mean ± s.d., $^{**}p < 0.01$, versus the control group; $^{\&}p < 0.05$ and $^{\&\&}p < 0.01$ versus group A; $^{\#}p < 0.05$ and $^{\#\#}p < 0.01$ versus group B.

## 3.2. Changes of the mRNA expression levels of antioxidant enzyme in the spleen

Consistent with the above results, a significant decrease in mRNA levels related to antioxidant enzymes was presented after *A. adenophora* treatment relative to the control. CAT mRNA levels were lower in groups B and C when compared with control and group A (figure 2*a*, $p < 0.01$), and GPx (figure 2*b*,

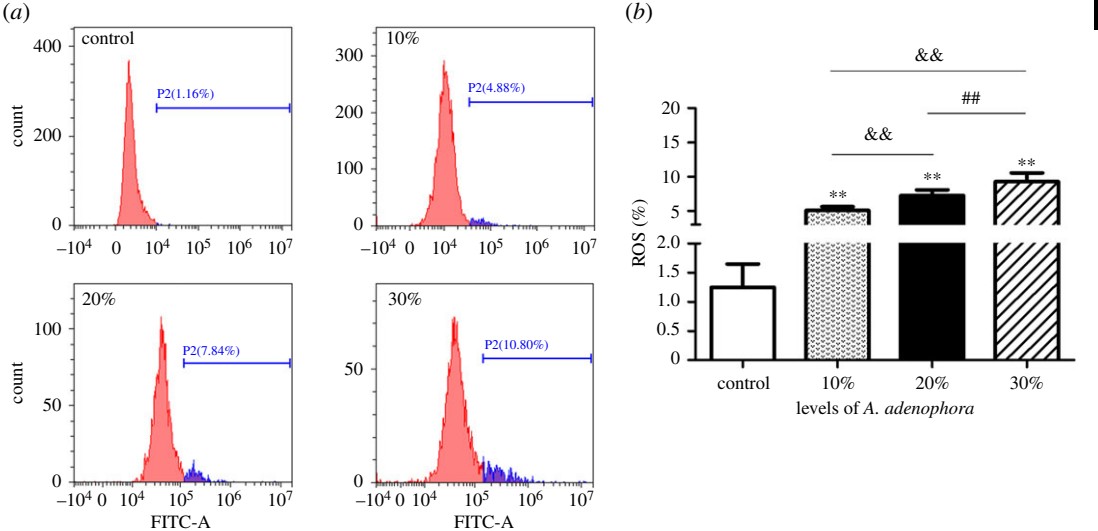

**Figure 3.** ROS levels in the *A. adenophora*-treated splenocytes. (*a*) Mice were treated with a different dosage of *A. adenophora* for 42 days. (*b*) Flow cytometry data showing ROS positive cells. The data are presented with the mean ± s.d., **$p < 0.01$, compared with the control group; $^{\&}p < 0.05$ and $^{\&\&}p < 0.01$ versus group A; $^{\#}p < 0.05$ and $^{\#\#}p < 0.01$ versus group B.

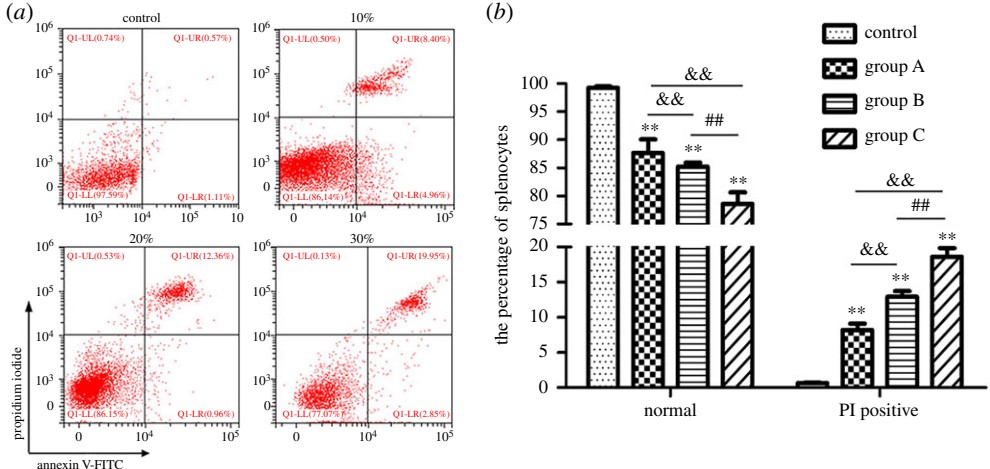

**Figure 4.** *Ageratina adenophora*-induced pyroptosis in splenocyte. (*a*) Scattergram showing percentage of pyroptotic splenocytes following annexin V-FITC/PI staining. (*b*) The percentage of PI+ cells increased significantly after *A. adenophora* was treated in a dose-dependent manner. All data are represented as mean ± s.d., **$p < 0.01$, compared with the control group; $^{\&\&}p < 0.01$ versus group A; $^{\#\#}p < 0.01$ versus group B.

$p < 0.01$), Mn-SOD (figure 2*c*, $p < 0.01$) and Cu–Zn SOD (figure 2*d*, $p < 0.01$) had declined in all *A. adenophora*-treated groups.

## 3.3. Detection of ROS production levels in the spleen

As shown in figure 3, the intracellular ROS levels increased significantly in the murine splenocytes of the three *A. adenophora* treatment groups at 42 days in a dose-dependent manner when compared between groups.

## 3.4. Detection of pyroptosis in splenocyte

Pyroptosis was evaluated by annexin V/PI staining. As shown in figure 4, the percentage of PI+ pyroptotic splenocytes was increased in a dose-dependent manner ($p < 0.01$) by comparing with all *A. adenophora*-administration groups. Furthermore, *A. adenophora* significantly upregulated the *in situ*

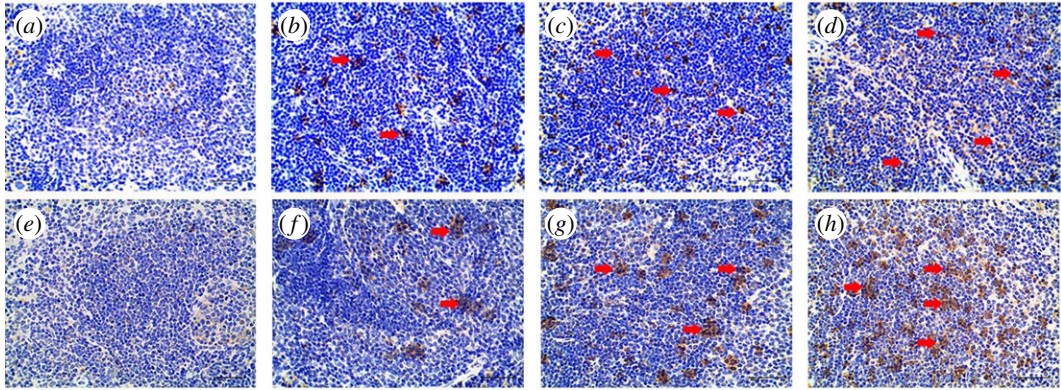

**Figure 5.** The expression of caspase-1 and IL-1β in the spleen after *A. adenophora* administration. (*a–d*) *In situ* caspase-1 expression in the control group (*a*), group A (*b*), group B (*c*) and group C (*d*). (*e–h*) *In situ* IL-1β expression in the control group (*e*), group A (*f*), group B (*g*) and group C (*h*). Caspase-1 and IL-1β positive regions are indicated by red arrows. Scale bars (*a–h*) = 40 μm.

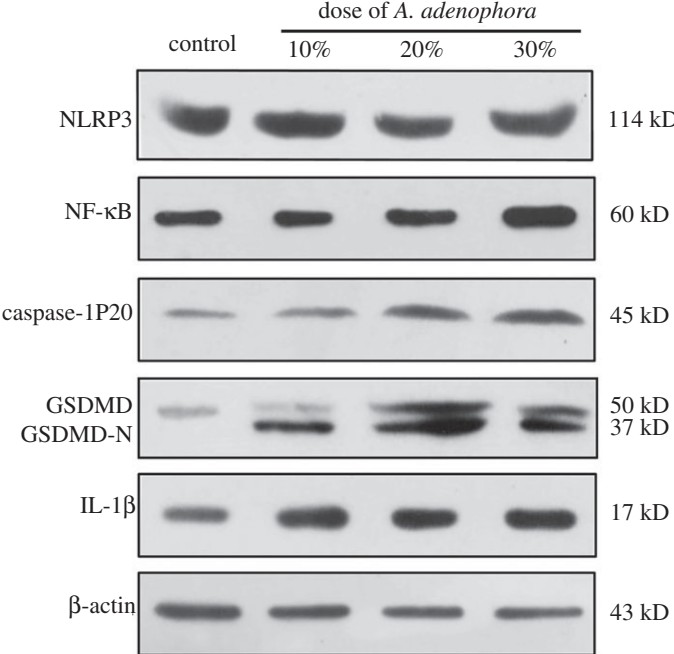

**Figure 6.** The change in protein expression associated with pyroptosis induced by *A. adenophora*.

expression of caspase-1 (figure 5*a–d*; electronic supplementary material, figure S1A) and IL-1β (figure 5*e–h*; electronic supplementary material, figure S1B) levels in the spleen in a dose-dependent manner, along with the elevation of caspase-1 activity (electronic supplementary material, figure S2A, $p < 0.01$) and IL-1β levels (electronic supplementary material, figure S2B, $p < 0.01$).

## 3.5. Changes of expression levels of protein associated with pyroptosis in the spleen

To test the effects of *A. adenophora* on pyroptosis pathway *in vivo*, we measured the protein levels of pyroptosis-related factors. As shown in the result, NRRP3 increased markedly following *A. adenophora* treatment. The protein level of caspase-1 was significantly increased by *A. adenophora* at the end of the experiment when compared with those in the control group. At the same time, the protein level of NF-κB was higher in groups B and C than those in the control group. In addition, we also measured the expression levels of the pyroptosis markers GSDMD. The GSDMD N-terminal (GSDMD-N) was only detected in the *A. adenophora*-treated groups. The results are shown in figure 6.

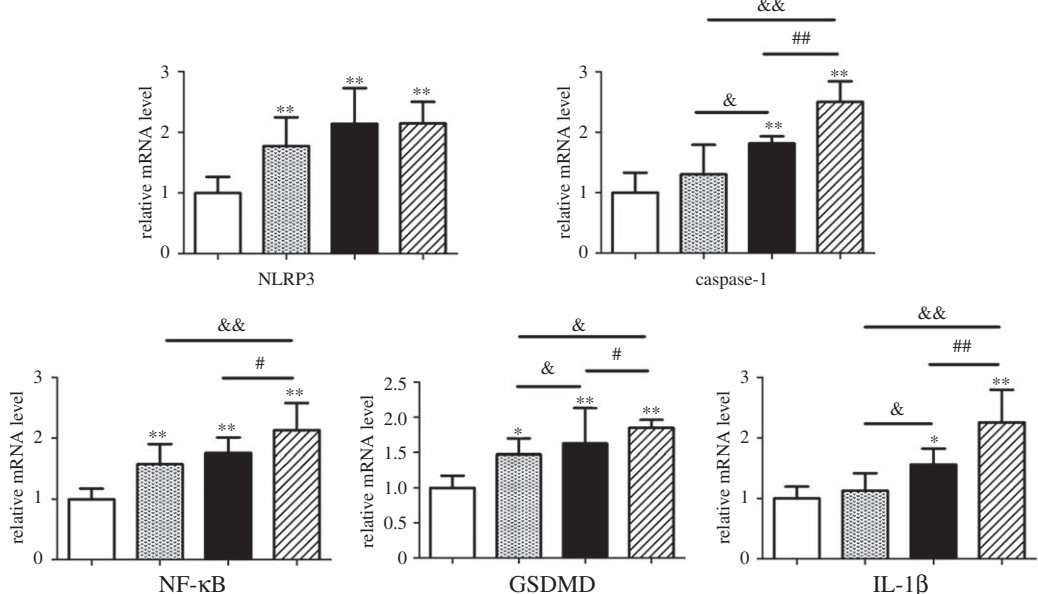

**Figure 7.** The change in mRNA expression related to pyroptosis induced by *A. adenophora*. Data are represented as mean ± s.d., *$p < 0.05$, **$p < 0.01$, compared with the control group; &$p < 0.05$ and &&$p < 0.01$ versus group A; #$p < 0.05$ and ##$p < 0.01$ versus group B.

## 3.6. Changes of mRNA expression levels of parameters associated with pyroptosis in the spleen

We further examine the mRNA level of pyroptosis-related factors. The results showed that the mRNA levels of NRRP3 were markedly increased ($p < 0.01$) following *A. adenophora* treatment. In addition, the mRNA expression levels of NF-κB were also increased ($p < 0.01$) in the *A. adenophora* treatment groups when compared between groups. In addition, the caspase-1 and IL-1β mRNA expression levels were upregulated ($p < 0.01$) in the *A. adenophora*-treated splenocytes in groups B and C but not in group A. The GSDMD mRNA expression levels were significantly increased ($p < 0.05$ or $p < 0.01$) by *A. adenophora*. These results are shown in figure 7.

## 4. Discussion

*Ageratina adenophora* is an invasive weed with widespread distribution [35] and reportedly causes hepatocytes apoptosis [10] and oxidative stress [13], as well as a systemic inflammatory response [6]. Since apoptosis is a non-inflammatory cell death programme [36], there is a strong possibility of another cell death mechanism in cells exposed to *A. adenophora*. We identified pyroptosis and oxidative damage as the underlying mechanisms of *A. adenophora*-induced splenic toxicity in mice.

In the present study, we found that *A. adenophora* enhanced splenocyte ROS levels in a dose-dependent manner, which coincided with MDA levels. ROS promotes lipid peroxidation [37], and MDA is a reliable indicator of this process [38]. Oxidative stress refers to the imbalance between ROS generation and the antioxidant response [39], which can trigger several pathological conditions [38]. SOD and CAT play an important role in scavenging ROS. GSH is also involved in eliminating ROS. In this study, *A. adenophora* decreased the splenic levels and activity of the antioxidant enzymes SOD, CAT and GPx, as well as GSH, a non-enzymatic antioxidant [40] that acts as the electron donor in the GPx-catalysed reduction of $H_2O_2$ to $H_2O$ [41]. In addition, to reveal the molecular changes of antioxidant enzyme activities, the mRNA expression levels of Mn-SOD, Cu–Zn-SOD, CAT and GPx were measured in this study. The results showed that *A. adenophora* decreased these antioxidant enzymes mRNA levels, which were consistent with the reduction of their activities. Taken together, *A. adenophora* increased ROS production and inhibited the antioxidant system in the spleen. The imbalance between ROS and antioxidant system results in oxidative stress, which damages the function of the spleen.

Spleen has a pivotal role in triggering immune regulation due to its anatomic location [42]. Immune cells recognize pathogens and other foreign antigens and recruit responder cells which initiate an immediate innate immune response [43]. Inflammation is a normal physiological response to an infection, irritant or other injury and helps in clearing the toxic debris [44]. However, sustained

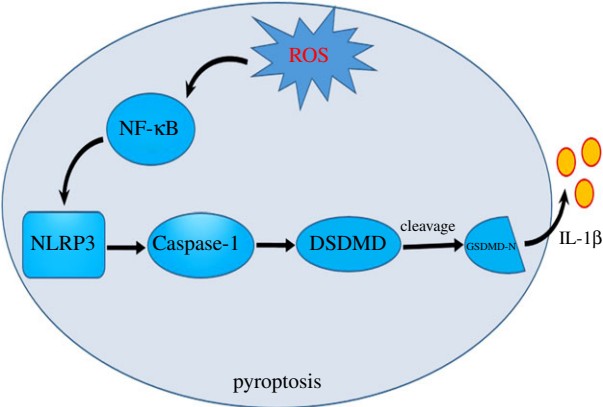

**Figure 8.** *Ageratina adenophora* causes spleen toxicity by inducing ROS and activating pyroptosis pathway in mice model.

inflammation can lead to massive tissue damage [45]. Some inflammatory diseases are mediated by oxidative stress [46], and ROS are known to induce the inflammatory response [20] that can lead to pyroptosis [22,23]. In this study, the percentage of pyroptotic splenocytes detected by flow cytometry was significantly increased after *A. adenophora* treatment. In order to reveal the pyroptosis mechanism caused by *A. adenophora*, we further detected the expression of a protein involved in the canonical pyroptosis signal pathway. Inflammasomes are assembled by sensing a variety of tissue injury signals. Diverse stimuli could promote the release of inflammatory cytokines, including IL-1β. NLRP3 is currently the best characterized one, which plays an important role in the immune system as a damage sensor. In addition, NLRP3 can also sense and be activated by ROS produced in the cell. It is known that GSDMD is a key pyroptosis executor [47]. GSDMD can be cleaved into GSDMD-N and GADMD-C by caspase-1. GSDMD-N promotes cell lysis and IL-1β release through forming pores by binding to the cell membrane. In this study, we found that the protein levels of NLRP3, NF-κB, caspase-1, GSMD and IL-1β were increased by inducing *A. adenophora*. Consistent with that, the mRNA of those proteins was also increased by *A. adenophora* in a dose-dependent manner. In addition, the activity of caspase-1 was also upregulated in *A. adenophora*-administration groups. GSDMD-N was only detected in *A. adenophora* treatment groups. This result further demonstrated the occurrence of pyroptosis induced by *A. adenophora*. Moreover, the IL-1β level was also increased by *A. adenophora*.

# 5. Conclusion

*Ageratina adenophora* impaired the spleen function in mice through oxidative stress damage and pyroptosis (figure 8). This result provided new insights for further understanding of the mechanism of splenic damage and toxicity induced by *A. adenophora*.

Ethics. This study was reviewed and approved by the Animal Care and Use Committee of Sichuan Agricultural University (SYXK2014-187). All animal operations, collection samples and procedures were carried out in accordance with the guidelines approved.
Data accessibility. The datasets supporting this article have been uploaded as part of the electronic supplementary material. And the raw material is deposited at Dryad: https://doi.org/10.5061/dryad.gf2np70 [48].
Authors' contributions. W.S. carried out the molecular laboratory work and drafted the manuscript. D.Y. performed the animal experiment and participated in data analysis. S.L. participated in the design of this work and data analysis. Z.R., Z.Z., J.D. and G.P. critically revised the manuscript. Y.H. conceived the study. C.Z. coordinated the study and helped draft the manuscript. All authors gave final approval for publication and agreed to be held accountable for the work performed therein.
Competing interests. The authors declare that they have no conflict of interests.
Funding. This work was supported by Science and Technology Support Program of Sichuan Province (No. 2015SZ0201).
Acknowledgements. Many thanks to Jinping Zhang, Fan Liu and Haiyang Zhou for their help in animal experiments.

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
