## [Reviewer comments · Royal Society Open Science]

Review History

RSOS-190127.R0 (Original submission)

Review form: Reviewer 1

Is the manuscript scientifically sound in its present form?

Yes

Are the interpretations and conclusions justified by the results?

Yes

Is the language acceptable?

Yes

Is it clear how to access all supporting data?

Yes

Do you have any ethical concerns with this paper?

No

Have you any concerns about statistical analyses in this paper?

Yes

Recommendation?

Major revision is needed (please make suggestions in comments)

Comments to the Author(s)

The manuscript by Sun et al., describes an experiment where 40 mice divided into groups of ten and then fed either control, 10%, 20%, or 30% *A. adenophora* in the form of a pelleted diet for 42 days. At the end of the study, the mice were sacrificed, spleens removed, and assays were performed.

Comments:

Introduction

General comment: What motivated the investigators to test pyroptosis? More detail would be helpful to understand why the work was performed. Perhaps this could be used to better organize the introduction.

Line 32, what regions worldwide?

Line 33, what is a grave impact? More detail please.

Line 36-40, please provide more detail about each species and what pathology this plant causes.

Lines 64-65, Due to the lack of detail I cannot assess the validity of the statement "that it is unclear whether the toxic effects of *A. adenophora* on splenocyte involve pyroptosis."

Materials and methods

General comment, for the sake of the reader who isn't an immunologist please try to limit your use of abbreviations or at least define them. Maybe a sentence or two about what a technique does. For example, what is cell quest software?

Line 79-81, How exactly was the plant material stored? Was a voucher specimen obtained and deposited in a herbarium? How exactly was this plant material incorporated in to a pellet? Was a chemical characterization of this plant material performed?

Line 152, what exactly is "mechanical method"?

Lines 154 - 158, The authors describe an ANOVA but no post-test. Why wasn't a post-test used?

Results

The authors should consider combining the results and discussion section to guide the reader through the data. This would allow for greater detail. For example, how exactly is NRRP3 an immune system damage sensor?

The authors mention dose-dependency, a chemical characterization of the plant material and specific concentrations of plant compounds would provide greater support to this statement.

Discussion

The authors set out to investigate if *A. adenophora* causes oxidative stress and pyroptosis. A figure illustrating the pyroptotic pathway and a point by point discussion of this pathway with regard to the experimental results would improve the discussion.

Review form: Reviewer 2**Is the manuscript scientifically sound in its present form?**

Yes

Are the interpretations and conclusions justified by the results?

Yes

Is the language acceptable?

No

Is it clear how to access all supporting data?

No

Do you have any ethical concerns with this paper?

No

Have you any concerns about statistical analyses in this paper?

No

Recommendation?

Major revision is needed (please make suggestions in comments)

Comments to the Author(s)

The authors present an interesting study of the spleen toxicity caused by *Ageratina adenophora*. This work provided a new insight into the mechanism of the toxicity of *A. adenophora* in a mice model. I agree to publish by Royal Society Open Science. But I have some comments:

1. Line 20: "Glutathione peroxidase" should be abbreviated to GPx. And abbreviations should be unified in line 94, 162 and 218.
2. The manuscript has some grammatical issues, and the authors should correct and improve them. The tenses of sentences should be consistent. Such as in lines 17-20, "the decrease..." should be corrected as "decreased the ..."
3. Lines 113: space should be between number and unit. Same errors should be corrected in Figure 6.
4. Line 75: The source format of reagent is generally expressed as "city, province and country", however, the sentence of line 75 lacks "province".
5. Line 123: replace " μ l" with " μ L".
6. Lines 190-191: What are "20% and 30% groups"?
7. Line 212: Please correct the expression of this sentence.
8. Although the manuscript is generally written clearly. Pay attention to minor errors. For Figure 5 caption, some words are misspelled, which is confusing to read.
9. In figure 3A, high resolution picture should be used so that the reader could see the data clearly.
10. There are some unscientific expressions. For instance, the "p" in line 199, 201 and 201 should be expressed in italics. The whole Ms should be revised.
11. Please modify the format of references according to the journal's requirements.

Review form: Reviewer 3 (PR Romão)

Is the manuscript scientifically sound in its present form?

Yes

Are the interpretations and conclusions justified by the results?

Yes

Is the language acceptable?

Yes

Is it clear how to access all supporting data?

Not Applicable

Do you have any ethical concerns with this paper?

No

Have you any concerns about statistical analyses in this paper?

No

Recommendation?

Accept with minor revision (please list in comments)

Comments to the Author(s)

The article is very interesting, and demonstrates the mechanisms involving the toxic effects of *Ageratina adenofora* in the spleen of treated mice. Some considerations should be examined. In the abstract the authors stated that the data indicate that *A. adenofora* impairs spleen function, however there is no evidences for this, despite the oxidative stress and pyroptosis. Maybe change the word indicate by suggest.

Please confirm the age of mice - 48 weeks?

Decision letter (RSOS-190127.R0)

21-May-2019

Dear Dr Hu,

The editors assigned to your paper ("*Ageratina adenophora* causes spleen toxicity by inducing oxidative stress and pyroptosis in mice") have now received comments from reviewers. We would like you to revise your paper in accordance with the referee and Associate Editor suggestions which can be found below (not including confidential reports to the Editor). Please note this decision does not guarantee eventual acceptance.

Please submit a copy of your revised paper before 13-Jun-2019. Please note that the revision deadline will expire at 00.00am on this date. If we do not hear from you within this time then it will be assumed that the paper has been withdrawn. In exceptional circumstances, extensions may be possible if agreed with the Editorial Office in advance. We do not allow multiple rounds of revision so we urge you to make every effort to fully address all of the comments at this stage. If deemed necessary by the Editors, your manuscript will be sent back to one or more of the original reviewers for assessment. If the original reviewers are not available, we may invite new reviewers.

When submitting your revised manuscript, you must respond to the comments made by the referees and upload a file "Response to Referees" in "Section 6 - File Upload". Please use this to document how you have responded to the comments, and the adjustments you have made. In

order to expedite the processing of the revised manuscript, please be as specific as possible in your response.

- Data accessibility

If you wish to submit your supporting data or code to Dryad (<http://datadryad.org/>), or modify your current submission to dryad, please use the following link:
<http://datadryad.org/submit?journalID=RSOS&manu=RSOS-190127>

- Competing interests

- Authors' contributions

- Acknowledgements

- Funding statement

on behalf of Dr Ryan Y Wong (Associate Editor) and Kevin Padian (Subject Editor)
openscience@royalsociety.org

Associate Editor's comments (Dr Ryan Y Wong):

Associate Editor: 1

Comments to the Author:

Dear Dr. Hu,

Your manuscript has been reviewed by 3 reviewers. While all reviewers agree that the work is interesting, several points of concerns were brought up. In particular there is a lack of description/details in some parts and the placement into context of broader field could be done more thoroughly. I recommend a major revision with a point-by-point response to each concern before considering the manuscript any further.

Comments to Author:

Reviewers' Comments to Author:

Reviewer: 1

Comments to the Author(s)

The manuscript by Sun et al., describes an experiment where 40 mice divided into groups of ten and then fed either control, 10%, 20%, or 30% *A. adenophora* in the form of a pelleted diet for 42 days. At the end of the study, the mice were sacrificed, spleens removed, and assays were performed.

Comments:

Introduction

General comment: What motivated the investigators to test pyroptosis? More detail would be helpful to understand why the work was performed. Perhaps this could be used to better organize the introduction.

Line 32, what regions worldwide?

Line 33, what is a grave impact? More detail please.

Line 36-40, please provide more detail about each species and what pathology this plant causes.

Lines 64-65, Due to the lack of detail I cannot assess the validity of the statement "that it is unclear whether the toxic effects of *A. adenophora* on splenocyte involve pyroptosis."

Materials and methods

General comment, for the sake of the reader who isn't an immunologist please try to limit your use of abbreviations or at least define them. Maybe a sentence or two about what a technique does. For example, what is cell quest software?

Line 79-81, How exactly was the plant material stored? Was a voucher specimen obtained and deposited in a herbarium? How exactly was this plant material incorporated in to a pellet? Was a chemical characterization of this plant material performed?

Line 152, what exactly is “mechanical method”?

Lines 154 – 158, The authors describe an ANOVA but no post-test. Why wasn’t a post-test used?

Results

The authors should consider combining the results and discussion section to guide the reader through the data. This would allow for greater detail. For example, how exactly is NRRP3 an immune system damage sensor?

The authors mention dose-dependency, a chemical characterization of the plant material and specific concentrations of plant compounds would provide greater support to this statement.

Discussion

The authors set out to investigate if *A. adenophora* causes oxidative stress and pyroptosis. A figure illustrating the pyroptotic pathway and a point by point discussion of this pathway with regard to the experimental results would improve the discussion.

Reviewer: 2

Comments to the Author(s)

The authors present an interesting study of the spleen toxicity caused by *Ageratina adenophora*. This work provided a new insight into the mechanism of the toxicity of *A. adenophora* in a mice model. I agree to publish by Royal Society Open Science. But I have some comments:

1. Line 20: “Glutathione peroxidase” should be abbreviated to GPx. And abbreviations should be unified in line 94, 162 and 218.
2. The manuscript has some grammatical issues, and the authors should correct and improve them. The tenses of sentences should be consistent. Such as in lines 17-20, “the decrease...” should be corrected as “decreased the ...”
3. Lines 113: space should be between number and unit. Same errors should corrected in Figure 6.
4. Line 75: The source format of reagent is generally expressed as “city, province and country”, however, the sentence of line 75 lacks “province”.
5. Line 123: replace “ μ l” with “ μ L”.
6. Lines 190-191: What are “20% and 30% groups”?
7. Line 212: Please correct the expression of this sentence.
8. Although the manuscript is generally written clearly. Pay attention to minor errors. For Figure 5 caption, some words misspelled, which is confusing to read.
9. In figure 3A, high resolution picture should be used so that the reader could see the data clearly.
10. There are some unscientific expression. For instance, the “p” in line 199, 201 and 201 should be expressed in italics. The whole Ms should be revised.
11. Please modify the format of references according to magazine’s requirements.

Reviewer: 3

Comments to the Author(s)

The article is very interesting, and demonstrates the mechanisms involving the toxic effects of *Ageratina adenophora* in the spleen of treated mice. Some considerations should be examined.

In the abstract the authors stated that the data indicate that *A. adenophora* impairs spleen function, however there is no evidences for this, despite the oxidative stress and pyroptosis. Maybe change the word indicate by suggest.

Please confirm the age of mice - 48 weeks?

Author's Response to Decision Letter for (RSOS-190127.R0)

See Appendix A.

RSOS-190127.R1 (Revision)

Review form: Reviewer 1

Is the manuscript scientifically sound in its present form?

Yes

Are the interpretations and conclusions justified by the results?

Yes

Is the language acceptable?

Yes

Do you have any ethical concerns with this paper?

No

Recommendation?

Accept as is

Comments to the Author(s)

Acceptable

Review form: Reviewer 2

Is the manuscript scientifically sound in its present form?

Yes

Are the interpretations and conclusions justified by the results?

Yes

Is the language acceptable?

Yes

Do you have any ethical concerns with this paper?

No

Recommendation?

Accept as is

Comments to the Author(s)

I am no another comments to this manuscript!

Decision letter (RSOS-190127.R1)

25-Jun-2019

Dear Dr Hu,

I am pleased to inform you that your manuscript entitled "Ageratina adenophora causes spleen toxicity by inducing oxidative stress and pyroptosis in mice" is now accepted for publication in Royal Society Open Science.

on behalf of Dr Ryan Y Wong (Associate Editor) and Kevin Padian (Subject Editor)
openscience@royalsociety.org

Reviewer comments to Author:
Reviewer: 1

Comments to the Author(s)
Acceptable

Reviewer: 2

Comments to the Author(s)
I have no other comments to this manuscript!

Appendix A

Dear editors,

Thank you for your kind comments on our manuscript entitled “*Ageratina adenophora* causes spleen toxicity by inducing oxidative stress and pyroptosis in mice”. Those comments are all valuable and very helpful for revising and improving our paper. We have studied the comments carefully and have revised the manuscript according to the reviews’ comments and suggestions. Revised portions are marked in red in the paper.

Responds to the reviews’ comments:

Reviewer 1:

Comments to the Author(s)

The manuscript by Sun et al., describes an experiment where 40 mice divided into groups of ten and then fed either control, 10%, 20%, or 30% *A. adenophora* in the form of a pelleted diet for 42 days. At the end of the study, the mice were sacrificed, spleens removed, and assays were performed. Comments:

Introduction

General comment: What motivated the investigators to test pyroptosis? More detail would be helpful to understand why the work was performed. Perhaps this could be used to better organize the introduction.

Response: We have detailed our motives to test pyroptosis in lines 62-66: “However, apoptosis is a form of cell death that avoid causing inflammation. This discrepancy might indicated a new cell death involved in the process of inflammation caused by *A. adenophora*. Pyroptosis is a pro-inflammatory form of regulated cell death, which regards as a general immune effector in in multiple cells. Although several studies have investigate the mechanism of toxicity induced by *A. adenophora*. The toxic effects of *A. adenophora* on splenocyte involve pyroptosis and oxidative damage still unknown.”

Line 32, what regions worldwide?

Response: We have revised in lines 27-28: “*Ageratina adenophora* (*A. adenophora*), also known as *Eupatorium adenophorum*, is a highly invasive weed species native to Mexico and Costa Rica, and has successfully invaded habitats across Europe, Oceania and Asia”.

Line 33, what is a grave impact? More detail please.

Response: We have revised in lines 29-31: “The southwest provinces of China are one of the worst affected regions, where *A. adenophora* reduces the biomass of other plants by altering the soil microbial community in the invaded areas. The invasion of *A. adenophora* to grassland indirectly leads to the reduction of the number of grazing animals and local plants, and the loss of biodiversity.”

Line 36-40, please provide more detail about each species and what pathology this plant causes.

Response: We have revised in lines 31-43: “In addition, *A. adenophora* is highly toxic to animals and affects multiple organs. For example, ingestion of this weed causes respiratory disease in horses. Intra-gastric administration of the freeze-dried leaf powder or methanol extract of *A. adenophora* resulted in multiple focal parenchymal necrosis and liver degeneration in mice. Rats fed with chow containing 25% (w/w) freeze-dried *A. adenophora* leaf powder developed jaundice, characterized by increased levels of plasma bilirubin, ALP, ALT and AST. Furthermore, rumination suspension and photosensitization have been caused in cattle. The toxic effects of *A. adenophora* ingestion on the liver, spleen and kidney of goat have also been demonstrated, with dose-dependent apoptosis and autophagy seen in goat tissues. A study have demonstrated that *A. adenophora* induced significant mice oxidative stress characterized by up-regulating mRNA levels of antioxidants, including superoxide dismutase (SOD), catalase (CAT) and glutathione (GSH). Consistent with this, our previous study has approved that $\geq 20\%$ dose of *A. adenophora* increased the liver weight and caused extensive inflammation, in addition to decreasing antioxidant activity, increasing the production of reactive oxygen species (ROS).”

Lines 64-65, Due to the lack of detail I cannot assess the validity of the statement “that it is unclear whether the toxic effects of *A. adenophora* on splenocyte involve pyroptosis.”

Response: We have revised this statement in lines 62-66 to clarify the purpose to investigate pyroptosis, which may be a new mechanism involved in the toxic effects caused by *A. adenophora*

Materials and methods

General comment, for the sake of the reader who isn't an immunologist please try to limit your use of abbreviations or at least define them. Maybe a sentence or two about what a technique does.

For example, what is cell quest software?

Response: In “Materials and methods” section, we have detailed the abbreviation of immunological related factors. Moreover, additional explanations are given for the techniques used in this study.

Line 79-81, How exactly was the plant material stored? Was a voucher specimen obtained and deposited in a herbarium?

Response: We have revised in lines 81-83: “The collected leaves were cleaned, grounded and screened to make dry power. The power was stored in shade condition with ambient temperature at 20 ± 2 °C.” *A. adenophora* specimen is not deposited in a herbarium but in our lab.

How exactly was this plant material incorporated in to a pellet?

Response: We have revised in lines 83-85: “For the preparation of 10%, 20% and 30% diet pellet, *A. adenophora* and mice feed were homogenized in water solution by the ratio of 1:9, 1:8 and 1:7, respectively. Then the diet was cast in the form of cylinders and dried at 27 °C for 48 h.”

Was a chemical characterization of this plant material performed?

Response: Plant material chemical characterization was not performed in this study. But the major active compound related to *A. adenophora* was researched by our lab in previous study. The detail information could be obtain from the papers entitled “Anti-NDV activity of 9-oxo10,11-dehydroageraphorone extracted from *Eupatorium adenophorum* Spreng in vitro”, “Clinical efficacy of 9-oxo-10,11-dehydroageraphorone extracted from *Eupatorium adenophorum* against *Psoroptes cuniculi* in rabbits” *et al.*

Line 152, what exactly is “mechanical method”?

Response: We have revised in line 132: “The spleen in each group was collected and ground into power by liquid nitrogen in a mortar.”

Lines 154-158, The authors describe an ANOVA but no post-test. Why wasn't a post-test used?

Response: It is helpful for understanding the results of this work by a post-test. And we have revised in Figure1-4 and 7. In addition, the descriptions of result have been corrected in Figure legends.

Results

The authors should consider combining the results and discussion section to guide the reader through the data. This would allow for greater detail. For example, how exactly is NLRP3 an immune system damage sensor?

Response: Lines 178-179, “To test the effects of *A. adenophora* on pyroptosis pathway in vivo, we measured the protein levels of pyroptosis-related factors.” was added in the result section. And we have detailed information in lines 218-223: “Inflammasomes are assembled by sensing a variety of tissue injury signals. Diverse stimuli could promote the release of inflammatory cytokines, including IL-1 β . NLRP3 is currently the best characterized one, which plays an important role in the immune system as damage sensor. In addition, NLRP3 can also sense and be activated by ROS produced in the cell. It is known that GSDMD is a key pyroptosis executor[48]. GSDMD can be cleaved into GSDMD-N and GADMD-C by caspase-1. GSDMD-N promotes cell lysis and IL-1 β release through forming pores by binding to cell membrane. Inflammasomes are assembled by sensing a variety of tissue injury signals. Diverse stimuli could promote the release of inflammatory cytokines, including IL-1 β . NLRP3 is currently the best characterized one, which plays an important role in the immune system as damage sensor. In addition, NLRP3 can also sense and be activated by ROS produced in the cell. It is known that GSDMD is a key pyroptosis executor[48]. GSDMD can be cleaved into GSDMD-N and GADMD-C by caspase-1. GSDMD-N promotes cell lysis and IL-1 β release through forming pores by binding to cell membrane.”

The authors mention dose-dependency, a chemical characterization of the plant material and specific concentrations of plant compounds would provide greater support to this statement.

Response: *A. adenophora* was ground to uniform power and then homogenized with feed to produce different levels of diet. The results showed that the degree of oxidative stress and pyroptosis aggravated with the increased levels of *A. adenophora*-administration. Plant material chemical characterization was not performed in this study, but could be found in our previous studies related to study on the biological properties of active compounds from *Ageratina adenophora* (Title: Euptox A induces G1 arrest and autophagy via p38 MAPK- and PI3K/Akt/mTOR-mediated pathways in mouse splenocytes).

Discussion

The authors set out to investigate if *A. adenophora* causes oxidative stress and pyroptosis. A figure illustrating the pyroptotic pathway and a point by point discussion of this pathway with regard to the experimental results would improve the discussion.

Response: A schematic diagram of *A. adenophora* causes oxidative stress and pyroptosis (Figure 8) was added in this revised version manuscript. It will be helpful for understanding pyroptotic pathway. And we have revised the discussion section in lines 218-223: “Inflammasomes are assembled by sensing a variety of tissue injury signals. Diverse stimuli could promote the release of inflammatory cytokines, including IL-1 β . NLRP3 is currently the best characterized one, which plays an important role in the immune system as damage sensor. In addition, NLRP3 can also sense and be activated by ROS produced in the cell. It is known that GSDMD is a key pyroptosis executor[48]. GSDMD can be cleaved into GSDMD-N and GADMD-C by caspase-1. GSDMD-N promotes cell lysis and IL-1 β release through forming pores by binding to cell membrane. Inflammasomes are assembled by sensing a variety of tissue injury signals. Diverse stimuli could promote the release of inflammatory cytokines, including IL-1 β . NLRP3 is currently the best characterized one, which plays an important role in the immune system as damage sensor. In addition, NLRP3 can also sense and be activated by ROS produced in the cell. It is known that GSDMD is a key pyroptosis executor[48]. GSDMD can be cleaved into GSDMD-N and GADMD-C by caspase-1. GSDMD-N promotes cell lysis and IL-1 β release through forming pores by binding to cell membrane.”

Special thanks to you for your good comments.

Reviewer: 2

Comments to the Author(s)

The authors present an interesting study of the spleen toxicity caused by *Ageratina adenophora*. This work provided a new insight into the mechanism of the toxicity of *A. adenophora* in a mice model. I agree to publish by Royal Society Open Science. But I have some comments:

1. Line 20: “Glutathione peroxidase” should be abbreviated to GPx. And abbreviations should be unified in line 94, 162 and 218.

Response: We have revised the abbreviation of “Glutathione peroxidase” in the manuscript in line 46, 96, 157, 165, 204-206, 534 and 539, respectively.

2. The manuscript has some grammatical issues, and the authors should correct and improve them. The tenses of sentences should be consistent. Such as in lines 17-20, “the decrease...” should be corrected as “decreased the ...”

Response: We have performed an extensive editorial revision on the manuscript to enhance expression, grammar constructions and word spelling. And we have revised the manuscript in lines 17-19: “*A. adenophora* significantly increased the levels of reactive oxygen species, and malondialdehyde, but decreased the antioxidants like catalase, superoxide dismutase, glutathione and glutathione peroxidase.”

3. Lines 113: space should be between number and unit. Same errors should corrected in Figure 6.

Response: We have revised in lines 113 and Figure 6.

4. Line 75: The source format of reagent is generally expressed as “city, province and country”, however, the sentence of line 75 lacks “province”.

Response: We have revised in line 76: “Biochemical assay kits were obtained from Nanjing Jiancheng Bioengineering Institute (Nanjing, Jiang Su, China).”

5. Line 123: replace “ μ l” with “ μ L”.

Response: We have replaced “ μ l” with “ μ L” in line 120: “In brief, 20 mg spleen tissue was lysised on ice in 200 μ L buffer for 30 min.”

6. Lines 190-191: What are “20% and 30% groups”?

Response: 20% and 30% groups represent group B and C, respectively. We have revised in lines 181-182: “At the same time, the protein level of NF- κ B was higher in the group B and C than those in the control group.”

7. Line 212: Please correct the expression of this sentence.

Response: We have revised the expression in lines 199-200: “In the present study, we found that *A. adenophora* enhanced splenocyte ROS levels in a dose-dependent manner, which were coincided with MDA levels.”

8. Although the manuscript is generally written clearly. Pay attention to minor errors. For Figure 5 caption, some words misspelled, which is confusing to read.

Response: We have revised the misspelled word “in sute” with “in situ” in lines 564-565: “In situ IL-1 β expression in the control group (E), group A (F), group B (G), group C (H).”

9. In figure 3A, high resolution picture should be used so that the reader could see the data clearly.

Response: We have adjusted the figure 3A resolution to 600 dpi.

10. There are some unscientific expression. For instance, the “p” in line 199, 201 and 201 should be expressed in italics. The whole Ms should be revised.

Response: We have corrected the “p” as “*p*” in the whole manuscript.

11. Please modify the format of references according to magazine’s requirements.

Response: For the reference format within the article, we are sorry for our incorrect citation format. And in this revised version, we have corrected all the reference formats to meet the requirements of Royal Society Open Science.

Special thanks to you for your good comments.

Reviewer: 3

Comments to the Author(s)

The article is very interesting, and demonstrates the mechanisms involving the toxic effects of *Ageratina adenofora* in the spleen of treated mice. Some considerations should be examined.

In the abstract the authors stated that the data indicate that *A. adenofora* impairs spleen function, however there is no evidences for this, despite the oxidative stress and pyroptosis. Maybe change the word indicate by suggest.

Response: Considering the reviewer’s suggestion, we have revised in lines 22-23: “These findings suggest that *A. adenophora* impairs spleen function in mice through oxidative stress damage and pyroptosis.”

Please confirm the age of mice - 48 weeks?

Response: We are very sorry for our incorrect writing “forty eight-week-old mice”. A total of 40 eight-week-old mice were used in this study. So we have revised in lines 87-88: “Forty 8-week-old female Kunming mice purchased from Experimental Animal Corporation of Dossy Biological Technology Company”

Thank you very much for your comments and suggestions.

Thanks again for the excellent and professional revision of our manuscript. Hopefully, we could have our article been considered of publication in this journal. Should there been any other corrections we could make, please feel free to contact us by email. My email is hychun114@163.com.

Yours sincerely,

Yanchun Hu

June 12th, 2019